# Morbidity, Clinical Course and Vaccination against SARS-CoV-2 Virus in Patients with Duchenne Muscular Dystrophy: A Patient Reported Survey

**DOI:** 10.3390/ijerph19010406

**Published:** 2021-12-30

**Authors:** Eliza Wasilewska, Agnieszka Sobierajska-Rek, Karolina Śledzińska, Sylwia Małgorzewicz, Ewa Jassem, Jolanta Wierzba

**Affiliations:** 1Department of Allergology and Pulmonology, Medical University of Gdansk, 80-211 Gdansk, Poland; ejassem@gumed.edu.pl; 2Department of Rehabilitation Medicine, Faculty of Health Sciences with Institute of Maritime and Tropical Medicine, Medical University of Gdansk, 80-211 Gdansk, Poland; sobierajska@gumed.edu.pl; 3Department of Internal and Pediatric Nursing, Faculty of Health Sciences with Institute of Maritime and Tropical Medicine, Medical University of Gdansk, 80-211 Gdansk, Poland; ksledzinska@gumed.edu.pl (K.Ś.); jolanta.wierzba@gumed.edu.pl (J.W.); 4Department of Clinical Nutrition, Medical University of Gdansk, 80-211 Gdansk, Poland; sylwiam@gumed.edu.pl

**Keywords:** COVID-19, SARS-CoV-2 virus, Duchenne muscular dystrophy, neuromuscular disease, rare diseases, vaccination against SARS-CoV-2, vaccination against COVID-19, anxiety, COVID-19 pandemic

## Abstract

*Background:* Patients with Duchenne muscular dystrophy (DMD) may be at higher risk of a severe course of COVID-19. The aim of the study was to evaluate: (1) the incidence and course of COVID-19 infection in DMD patients; (2) the vaccination status of DMD patients; and (3) COVID-19 related anxiety among DMD families. *Materials and Methods:* The study was conducted during an online symposium for DMD patients and their families. All participants (DMD families; *n* = 150) were asked to fill in the online survey with questions about COVID-19 infection history, vaccination against SARS-CoV-2 and anxiety during pandemic. *Results:* 53 DMD patients filled in the survey. Five (9.43%) were COVID-19 positive with mild symptoms of respiratory infection and anosmia; 23 (42.6%) were vaccinated, but in almost 20% of DMD families, none of the family members was vaccinated. Respondents revealed anxiety related both to the vaccination procedure and to COVID-19 infection (complications after infection 93.6%, death 62.4% respondents). Changes in health care system organization also aroused concern among participants (85.3%). *Conclusion:* The course of the COVID-19 infection in DMD patients was mild. Not enough patients with DMD and their families are vaccinated. Education about the management of COVID-19 infections and the vaccination procedure for DMD patients is needed and expected.

## 1. Introduction

Since December 2019, the coronavirus disease 2019 (COVID-19) caused by the severe acute respiratory syndrome virus (SARS-CoV-2) has rapidly spread all over the world. In Poland, the COVID-19 pandemic was officially announced by the Ministry of Health on 13 March 2020 [1]. Recently, the number of new cases and deaths in Poland due to COVID-19 has rapidly increased to a total number of more than 2,330,000 cases and 78,688 deaths reported to date (17 November 2021) [2].

In the literature, several risk factors of a severe clinical course of COVID-19 have been described: obesity, chronic respiratory tract diseases, hypertension, chronic renal illnesses, diabetes and malignancy [3,4]. Patients with Duchenne muscular dystrophy (DMD) due to progressive muscular weakness, which causes respiratory failure, may be at higher risk of a severe course of COVID-19. *DMD* is the most common, progressive, irreversible muscular dystrophy inherited in an X-linked recessive mode. Mutations in the *DMD* gene disrupt the process of dystrophin protein production leading to a continuous decrease in muscular tension and power and resulting in loss of ambulation, respiratory failure and cardiomyopathy [5,6,7]. To prolong the independent mobility period, as well as pulmonary and cardiac function, steroid treatment is provided. Despite beneficial effects, prednisone or deflazacort may be associated with many side effects such as increased appetite, causing obesity, or immunosuppression leading to a severe course of the COVID-19 disease. Moreover, chronic steroid use may diminish the vaccination response, making patients more vulnerable to COVID-19 [8]. The COVID-19 pandemic has raised many questions regarding the appropriate management of infected patients, not only those who were previously healthy, but also and especially those suffering from chronic illnesses. Patients with neuromuscular disorders may be particularly at risk, due to the possible deterioration of respiratory function caused by muscle weakness.

During the pandemic, patients with disabilities can feel even more isolated and abandoned due to restrictions in personal contact, and they experience more difficulties in obtaining adequate multidisciplinary care. Children with DMD and their families are at risk of emotional imbalance with anxiety or depression affecting their everyday lives [9].

Currently, there are few studies regarding information on the clinical course and management of COVID-19 infections in patients with DMD, and there is not enough information regarding public health policies for this particular group of patients with respect to vaccinations and related concerns [10].

The aim of this study was to evaluate: 1. the incidence, the course and the outcomes of COVID-19 infection in DMD patients; (2) the vaccination status of DMD patients; and (3) the anxiety related to COVID-19 among families with DMD.

## 2. Materials and Methods

### 2.1. Study Design

In this multicenter, descriptive observational study the incidence and the course of COVID-19 infection and SARS-CoV-2 vaccination status among DMD patients were evaluated.

The study was conducted as part of the Multidisciplinary Care Program for Patients with Duchenne Muscular Dystrophy at the Rare Disease Centre (RDC), University Clinical Centre, Medical University of Gdansk, Poland. The University Clinical Centre is a member of the TREAT NMD Alliance Neuromuscular Network.

Approval for the study was obtained from The Committee of Ethics no. NKBBN/260/2021, which conformed to the principles embodied in the Declaration of Helsinki.

### 2.2. Participants

The study included male DMD patients diagnosed on the basis of the presence of clinical symptoms and genetic test results and/or muscle biopsy results [5]. All patients, together with their families, are members of the Parent Project Muscular Dystrophy Foundation in Poland [11]. Patients were recruited during the 5th International Symposium “Possibilities of supporting the development of people with rare diseases—Duchenne Muscular Dystrophy and Other Muscular Dystrophies” organized by the Rare Disease Centre and the Parent Project Muscular Dystrophy Foundation, online, on 19–20 November 2021 [12].

The topics of the conference concerned many aspects of the multidisciplinary care of DMD and was primarily dedicated to DMD patient and their families, caregivers and medical professionals involved in DMD care.

### 2.3. SARS-CoV-2 Survey

All participants (DMD families; *n* = 150) were asked to fill in the online anonymous survey (Google Forms), which evaluated the incidence and the course of COVID-19 among DMD families, as well as questions related to COVID-19 vaccinations. The responses were opened at 06:00, 12 November 2021 and closed at 23:59, 27 November 2021.

The questionnaires were completed in Polish through an online survey platform via the link placed on the conference website [13].

The structured questionnaire consisted of questions that covered several areas: (1) demographic data and clinical status; (2) COVID-19 incidence; (3) vaccination against SARS-CoV-2 virus; (4) anxiety during the COVID-19 pandemic.

(1)The demographic data consisted of information regarding age, weight, height, daily activity (kindergarten, school, home school, work) and the number of household members. Clinical course was assessed by evaluating ambulation status using the Vignos Scale (VS), and Brooke scale (BS). Both of these scales are commonly used to determine functional dependence in neuromuscular diseases. The VS allows the monitoring of the course of the disease and focuses on functional activities, mainly of the lower limbs, in which deterioration is considered an important milestone in the progression of the disease. The scores on the VS range from 1 to 10; “1” means that the subject can walk and climb stairs without assistance, while “10” means that the subject is confined to bed [14]. The BS assesses upper limb functional status with the 6-point Brooke scale, in which “1” means that the patient can abduct their arms in a full circle until they touch above their head, while “6” means that the patient has no useful function of the hands [15].

Function of the respiratory system was determined on the basis of saturation and the need to use non-invasive ventilation (NIV). Data regarding steroid therapy (yes/no) and the daily dosage (mg) of steroids was also collected.

(2)In the COVID-19 incidence section, participants were asked about their history of COVID-19 (yes/no), possible clinical symptoms, as well as duration and complications of COVID-19 infection.(3)Participants were also asked about vaccinations against the SARS-CoV-2 virus and the possible reasons for refusal.(4)In the anxiety section, participants were asked about emotions experienced in relation to (a) the possibility of being infected with COVID-19 (fear of death, complications, worsening of the course of DMD) and (b) changes in healthcare service (deterioration of medical professional availability and care). A five-level Likert scale (one-strongly disagree, five-strongly agree) was used to appropriately obtain unified questions and answers. The anxiety questionnaire was initially used and validated for assessing fear of COVID-19 among the Polish population of parents of children with Pulmonary Arterial Hypertension [16].

### 2.4. Statistical Analysis

The results of the statistical analysis were expressed as mean and standard deviation (SD) or median and first and third quartile (Q1; Q3) with 95% confidence intervals. The statistical analysis was carried out with Statistica 13.3 (StatSoft, Krakow, Poland).

## 3. Results

### 3.1. DMD Cohort Characteristics

During open response time, only 53 (35%) of the 150 conference participants responded to the survey. The clinical characteristics of the DMD cohort are shown in Table 1. The median age of the DMD patients was 12.02 years (the range was from 2 to 32 years); the majority of them—40 (75.5%)—used steroid therapy. In total, 22 patients were non-ambulatory; the remaining nine were able to ambulate independently or with support. There were two people without any useful function of both upper and lower limbs (VS-9, BS-6).

The majority of participants were school-aged children: thirty-four attended school (64.2%), nine participated in homeschooling (17%), seven patients attended to kindergarten (13.2%) and one respondent attended university.

### 3.2. COVID-19 Infection

A COVID-19 infection was declared by five (9.43%) of the DMD participants (see Table 2). The median age of infected DMD participants was 9 years (aged 3.5 to 23 years). The major symptoms of COVID-19 infection presented in DMD patients were as follows: sore throat, headache, muscle pain, fever, cough and anosmia. None of the infected persons required NIV or hospitalization, and none declared serious complications from COVID-19.

### 3.3. Vaccination against SARS-CoV-2 Virus

In the study group, 23 (42.6%) patients were vaccinated against COVID-19; the mRNA vaccine (COMIRNATY; Pfizer, Inc., and BioNTech, Mainz, Germany) was given to the deltoid muscle of the upper limb. The other 31 (57.4%) were not vaccinated. All unvaccinated participants and their families were aware of the possibilities regarding time and place for possible vaccinations. Due to age restrictions, nine boys (30%) were not vaccinated, because they were less than 12 years old. The main reason for the vaccination refusal of the remaining 22 DMD patients was fear of the vaccination procedure and of complications (70%); five of these boys and their parents were not vaccinated.

Details of the vaccinated and non-vaccinated group are shown in Table 3.

None of the DMD patients with COVID-19 history were vaccinated before infection. After the infection, one person decided to vaccinate; three respondents were not vaccinated due to young age (<12 years).

### 3.4. Anxiety during the COVID-19 Pandemic

Majority of participants (48/53) answered the questions regarding anxiety about infection and its consequences. Most concerns were related to fear of complications (93.6%), mainly death due to infection (62.4%); secondly, participants were afraid of changes in health care system organization (85.3%); and, about the impact of COVID-19 on disease course in DMD patients (75%). Detailed results of all anxiety issues during the pandemic are presented in Table 4.

## 4. Discussion

In our study, the frequency and clinical presentation of COVID-19 infectionand the vaccination status of patients with DMD, aged 2–32, of Caucasian race, are presented.

Only 9.43% of evaluated DMD patients were infected with COVID-19. Most of them presented mild symptoms of the infection, such as low-grade fever, headache, muscle pain, anosmia or sore throat. None of them was admitted to the hospital, nor was non-invasive/invasive ventilation needed, and all of the patients fully recovered, without major complications.

Less than 50% of the analyzed group of DMD patients was vaccinated against COVID-19; none of the infected patients was vaccinated. Apart from a lower age (less than 12 years old; 30% of unvaccinated patients), the main reason of refusal was anxiety (70%).

Anxiety was usually associated with not having adequate knowledge about the impact of COVID-19 on the course of DMD, vaccine function and its adverse reactions, and changes in health care system organization during the pandemic.

In Poland, current statistics for new COVID-19 cases and deaths do not include age criteria; it is not possible to assess the incidence rate of COVID-19 infection among children. Therefore, we are unable to reliably compare our results with our international colleagues.

Levine et al. reported that 6.1% of children with DMD were infected by the SARS-CoV-2 virus in Israel, which was less than the percentage of COVID-19 infection among the general pediatric population in this country. The authors described the experience of six DMD patients and one BMD patient diagnosed with COVID-19, out of the 116 patients who remained under their care [8]. Only two were admitted to the hospital due to the dyspnea, chest pain, fever or headache; one needed non-invasive ventilation; however, in the end, all recovered without major complications. The mean age of the patients was 14 years old; and five of them were non-ambulatory and five of them were obese; all of them were infected by a first-degree family member.

A Spanish center also shared the experience of 29 children with neuromuscular disorders infected by SARS-CoV-2—four boys with DMD and one BMD patient [17]. The majority of them were described as asymptomatic or mild cases, while only 10% were described as moderate. Only four patients were hospitalized, and all of them were suffering from SMA.

Similarly, our study showed that the clinical symptoms of COVID-19 infection were not very severe, and all patients recovered without any significant sequelae. It confirms previously reported experience of a mild or asymptomatic course of the disease in the majority of children with DMD [8,17].

The same findings are observed in older DMD patients with an advanced stage of the disease; our adult DMD patients infected with COVID-19 presented only with low-grade fever and anosmia. In the UK, Quinlivan et al. reported their findings in seven older DMD patients (aged 17–26 years) diagnosed with COVID-19 [18]. All of them presented a benign course of the infection. Two of them remained asymptomatic, two presented with anosmia and reduced sensation and the remaining three complained of a transient fever with a cough, runny nose or sore throat. All of them experienced full recovery.

It is important to underline that, in our study, we reported more than 50% of DMD patients who were not vaccinated, with no family member vaccinated in the case of 20% of the respondents. Although most patients were infected by family members, we still noticed one family in which nobody was vaccinated even after a COVID-19 infection in a boy with DMD. The main reason for non-vaccination, apart from younger age, was anxiety related to unknown tolerance for the vaccine and the possibility of adverse reactions.

Interestingly, most of the respondents also reported anxiety in relation to the presence of possible complications of COVID-19 infection. The majority of respondents also reported a lack of sufficient knowledge about the impact of COVID-19 on the course of DMD. Although the pandemic started almost 2 years ago, there are still not enough data regarding the course of COVID-19 in DMD patients. No common management recommendations for DMD patients in the COVID-19 pandemic have been published for either patients or doctors.

In the US, Veerapandiyan et al. recommended continuing corticosteroid therapy and considering additional stress dosing during illness or hospitalization. Similarly, exon skipping therapy or cardiology medications (ACE-inhibitors or angiotensin receptor blockers) should be continued after a discussion of the risk-benefit ratio. As is recommended in the international guidelines for care of DMD patients with deteriorating respiratory function, oxygen should be used only together with ventilatory support. Further multidisciplinary care management should be offered, while adjusting to patients’ needs and current epidemiological status [6,10].

Moreover, our study showed that patients are afraid of unpredictable changes in health care system organization, as well as of changes in annual assessment schedule.

Current WHO recommendations highlight the role of telemedicine [19]. In Saudi Arabia, Bamaga et al. advised DMD patients to proceed with virtual clinic visits, if possible, to avoid unnecessary contact with other patients who are probably infected [20]. In our previous study, we presented the possibilities and benefits of home e-monitoring of the pulmonary function in DMD patients [21].

Ideally, rehabilitation and physiotherapy should be continued in a home-based setting, if possible. Although traditional rehabilitation and physiotherapy with real patient-provider contact is the optimal way of prolonging the satisfactory physical condition of DMD patients, during the COVID-19 pandemic, such exercises were performed less often, if at all. To help DMD patients overcome this obstacle, many multidisciplinary care centers proposed telemedicine tools to continue proper physiotherapy management for their patients. In Poland, Sobierajska-Rek et al. showed that, with the guidance of physiotherapists (via online communication or video), patients with caregivers’ help can continue home based rehabilitation. Online videos, instructions and video guidelines are more acceptable to parents and caregivers of patients with DMD than live workshops [22].

To summarize, we would like to conclude that the course of the declared COVID-19 infections among the studied DMD patients was mild and was not related to any serious complications. The anxiety of patients and their caregivers was usually associated with inadequate knowledge of the impact of COVID-19 on the course of DMD, with unknown vaccine functions and with possible adverse reactions. We conclude that more education is needed in relation to the benefits of vaccination in chronically ill children.

A limitation of the study was the relatively small group of subjects. Moreover, the assessment was only based on respondents’ declarations; thus, we were not able to avoid bias in appropriate assessment of COVID-19 symptoms. Additionally, asymptomatic participants were not tested for SARS-CoV-2, limiting our knowledge about the number of asymptomatic DMD patients.

However, we still consider this study to be interesting, as it showed the incidence and clinical course of COVID-19 infections among DMD patients in Poland.

To the best of our knowledge, this is the first report showing the incidence and circumstances of COVID-19 vaccinations among DMD patients.

### 4.1. Simple Summary

The SARS-CoV-2 virus pandemic has raised many questions concerning the appropriate management of infected patients, especially for those suffering from chronic illnesses. Patients with neuromuscular disorders may be particularly at risk, due to possible respiratory function deterioration caused by muscle weakness.

Our results suggest that the course of COVID-19 in children with DMD was not as severe as expected. Although children with DMD appear to be more susceptible to COVID-19, less than half of them were vaccinated. In addition, many families of boys with DMD remain unvaccinated. The main reason for the unwillingness to receive vaccination is anxiety related to possible side effects and inadequate information regarding the benefits of vaccination.

### 4.2. Clinical Implications

This study shows the need for more education for DMD patients and their families concerning COVID-19 infections in chronic illness patients and the benefits of vaccination.

## 5. Conclusions

The course of COVID-19 infection in DMD patients was mild. Not enough patients and their families are vaccinated. Education about the management of COVID-19 infections and about vaccination procedures for DMD patients is needed and expected.

## Figures and Tables

**Table 1 ijerph-19-00406-t001:** Clinical characteristics of DMD cohort.

	DMD Participants *n* = 53
Age y (Mean, SD) [95% CI]	12.02 (5.5) [10.5–13.5]
BMI kg/m^2^ (Mean, SD) [95% CI]	21.04 (5.3) [19.5–22.5]
Ambulatory (*n*, %)	31 (58.5%)
VS (Median, Q1; Q3) [95% CI]	4 (2; 9) [4–6]
BS (Median, Q1; Q3) [95% CI]	2 (1; 2) [1–2]
**Comorbidities (*n*, %)**	**16 (30.2%)**
Cardiomyopathy	3 (5.7%)
Cognitive delay	1 (1.9%)
Autistic spectrum disorders	3 (5.7%)
Hypothyroidism	6 (11.3%)
Others	3 (5.7%)
Steroid therapy	43 (81.1%)
ACE inhibitors	38 (70.4%)
NIV	2 (3.8%)
Family members at home (Median, Q1; Q3)	4 (3; 4) [3–4]
**Family members vaccinated (*n*,%)**	
All family members at home	19 (35.8%)
Only mother	5 (9.4%)
Mother and father	14 (26.4%)
Only father	3 (5.7%)
Nobody	10 (18.9%)
Answer not given	2 (3.8%)

BMI, body mass index; VS, Vignos scale; BS, Brook scale; NIV, non-invasive ventilation.

**Table 2 ijerph-19-00406-t002:** Clinical characteristics of infected SARS-CoV-2 virus patients.

No. Patient	1	2	3	4	5
Age y	21	6	3.5	9	23
BMI kg/m^2^	22.79	20.19	13.88	17.88	35.45
Ambulation	no	yes	yes	yes	no
VS	9	1	1	2	8
BS	6	1	1	2	5
Comorbidities	Asperger syndrome	None	Celiac disease	Hypothyroidism	Cardio-myopathy
Steroid therapy	yes	Yes	yes	yes	Yes
ACE inhibitors	yes	Yes	yes	yes	Yes
Saturation NIV	74%	97%	98%	95%	95%
	no	no	no	no	no
Source of COVID-19 infection	family	School	family	not known	Family
Clinical symptoms	fever	sore throat, headache, muscle pain, anosmia	fever	sore throat, muscle pain	anosmia
Complications	no	no	no	no	no
Infection date	Dec 2020	Nov 2020	Nov 2020	Feb 2021	Dec 2020
Vaccination date	Jun 2021	No	no	no	No
Family vaccination date	Jun 2021	Jun 2021	no	Jul 2021	Oct 2021
Family members at home (*n*)	4	5	4	6	4

BMI, body mass index; VS, Vignos scale; BS, Brook scale; NIV, non-invasive ventilation.

**Table 3 ijerph-19-00406-t003:** Details regarding vaccination against SARS-CoV-2.

	Vaccinated	Not Vaccinated Due To Age	Not Vaccinated
Age y (Median, Q1; Q3) [95% CI]	16, (12; 16) [12.8–17.5]	8, (5; 11) [6.4–9.6]	12, (10; 13) [8.7–14.4]
VS (Median, Q1; Q3) [95% CI]	9, (3; 9) [5–8]	2, (2; 4) [2–5]	4, (1; 9) [2–7]
BS (Median, Q1; Q3) [95% CI]	1, (1; 5) [1–3]	1, (1; 2) [1]	2, (1; 2) [1–3]
Number of family members at home (Median, Q1; Q3) [95% CI]	3, (3; 4) [3–4]	4, (2; 5) [3–4]	4, (3; 4) [3–4]

VS, Vignos scale; BS, Brook scale.

**Table 4 ijerph-19-00406-t004:** Detailed results of causes of anxiety during pandemic.

	Strongly Disagree	Disagree	No Opinion	Agree	Strongly Agree
COVID-19 Infection	4 (8.3%)	11 (22.9%)	2 (4.1%)	15 (31.2%)	16 (33.3%)
Complications after infection	1	1	1	26 (54.1%)	19 (39.5%)
Death due to infection	6 (12.5%)	11 (22.9%)	1	11 (22.9%)	19 (39.5%)
Changes in health care system organization	2 (4.1%)	4 (8.3%)	1	20 (41.6%)	21 (43.7%)
Changes in annual assessments schedule	4 (8.3%)	8 (16.6%)	0	20 (41.6%)	16 (33.3%)
Lack of knowledge of how COVID-19 affects DMD	4 (8.3%)	7 (14.6%)	1	18 (37.5%)	18 (37.5%)
School system changes	17 (35.4%)	10 (20.8%)	1	11 (22.9%)	9 (18.7%)

## Data Availability

The data presented in this study are available on request from the corresponding author. Full data are not publicly available due to privacy restrictions.

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
