# Peer review of "Morbidity, Clinical Course and Vaccination against SARS-CoV-2 Virus in Patients with Duchenne Muscular Dystrophy: A Patient Reported Survey"

_ijerph, 2021, doi:10.3390/ijerph19010406_

Round 1

Reviewer 1 Report

The study is only descriptive, but the data reported are of some interest in its field. The sample size is limited, but overall acceptable, giving the considered disease.

Some clarifications:

how was the questionnaire created and eventually validated? How was its structure thought? Were other questionnaires used for its creation?

Anxiety status: a general anxiety questionnaire (i.e GAD-7 or similar) to be accompanied to the questions about anxiety for COVID could be of interest.

What about the symptoms of patients after COVID vaccination?

Author Response

Thank you for your valuable comments. We revised the manuscript and presented our answers below according to the Reviver’s suggestions.

How was the questionnaire created and eventually validated? How was its structure thought? Were other questionnaires used for its creation?

Anxiety status: a general anxiety questionnaire (i.e GAD-7 or similar) to be accompanied to the questions about anxiety for COVID could be of interest.

The anxiety scale was initially used and validated for assessing fear of COVID-19 among the Polish population of parents of children with Pulmonary Arterial Hypertension by Kwiatkowska J et al.

Kwiatkowska J, Meyer-Szary J, Mazurek-Kula A, Zuk M, Migdal A, Kusa J, et al. The Impact of COVID-19 Pandemic on Children with Pulmonary Arterial Hypertension. Parental Anxiety and Attitudes. Follow-Up Data from the Polish Registry of Pulmonary Hypertension (BNP-PL). J Clin Med 2021;10:1640. https://doi.org/10.3390/jcm10081640.

What about the symptoms of patients after COVID vaccination?

Unfortunately, we did not investigate. Thank you for your valuable insight. We plan to investigate this issue in the future when a larger group of patients with DMD will be vaccinated.

Reviewer 2 Report

I have some minor comments in this research report which are given below.

Study design should be classified as ‘descriptive observational study’ under the methods section.

Kindly do not call ‘subjects’ rather use ‘study participants’ or just ‘patients’ throughout the manuscript.

When reporting incidence and other proportion rates, authors should add 95% confidence intervals as interval estimates are better representation of data compared to point estimate.

For assessing anxiety levels, how questions were selected? Was it validated scale or authors prepared questions? If it was a validated scale to assess, please add detail with citation. If it was prepared by authors, then add reliability and face validity information of anxiety question.

Table 1 and 3, Authors should present the 1st and 3rd quartile values rather than the difference between these two values.

Gender information is missing in the table 1. Kindly also add the unit of measurements for age, BMI, VS, and BS.

There is a typo in the line ‘Only 9,43% of evaluated DMD patients were infected with COVID-19.’ as percent value should be written as ‘9.43%’ format that used throughout the manuscript.

Author Response

Thank you for your valuable comments. We revised the manuscript and presented our answers below according to the Reviver’s suggestions.

Study design should be classified as ‘descriptive observational study’ under the methods section.

We added the information to the Method section.

Kindly do not call ‘subjects’ rather use ‘study participants’ or just ‘patients’ throughout the manuscript.

Changes were made according to the Reviewer's suggestion.

When reporting incidence and other proportion rates, authors should add 95% confidence intervals as interval estimates are better representation of data compared to point estimates.

Changes were made according to the Reviewer's suggestion.

For assessing anxiety levels, how questions were selected? Was it validated scale or authors prepared questions? If it was a validated scale to assess, please add detail with citation. If it was prepared by authors, then add reliability and face validity information of anxiety question.

The anxiety questionnaire was initially used and validated for assessing fear of COVID-19 among the Polish population of parents of children with Pulmonary Arterial Hypertension by Kwiatkowska J et al.

Kwiatkowska J, Meyer-Szary J, Mazurek-Kula A, Zuk M, Migdal A, Kusa J, et al. The Impact of COVID-19 Pandemic on Children with Pulmonary Arterial Hypertension. Parental Anxiety and Attitudes. Follow-Up Data from the Polish Registry of Pulmonary Hypertension (BNP-PL). J Clin Med 2021;10:1640. https://doi.org/10.3390/jcm10081640.

Table 1 and 3, Authors should present the 1st and 3rd quartile values rather than the difference between these two values.

Author response: Q1 and Q3 were added to the values in  Table 1 and Table 3

Gender information is missing in the table 1. Kindly also add the unit of measurements for age, BMI, VS, and BS.

Changes were made according to the Reviewer's suggestion.

We have not provided the units for VS and BS due to the fact that these are scales used for staging DMD patients' general functional condition. Please kindly see both scales below:

Vignos scale:

  1. Walks and climbs stairs without assistance
  2. Walks and climbs stairs with aid of railing
  3. Walks and climbs stairs slowly with aid of railing (over 25 seconds for 8 standard steps)
  4. Walks unassisted and rises from chair but cannot climb stairs
  5. Walks unassisted but cannot rise from chair or climb stairs
  6. Walks only with assistance or walks independently with long leg braces
  7. Walks in long leg braces but requires assistance for balance
  8. Stands in long leg braces but unable to walk even with assistance
  9. Is in a wheelchair
  10. Is confined to bed

Vignos PJ, Spencer GE, Archibald KC. Management of progressive muscular dystrophy in childhood. JAMA 1963;184:89–96. https://doi.org/10.1001/jama.1963.03700150043007.

Brooke scale:

  1. Starting with arms at the sides, the patient can abduct the arms in a full circle until they touch above the head
  2. Can raise arms above head only by flexing the elbow (shortening the circumference of the movement) or using accessory muscles
  3. Cannot raise hands above head, but can raise an 8-oz glass of water to the mouth
  4. Can raise hands to the mouth, but cannot raise an 8-oz glass of water to the mouth
  5. Cannot raise hands to the mouth, but can use hands to hold a pen or pick up pennies from the table
  6. Cannot raise hands to the mouth and has no useful function of hands

Brooke MH, Fenichel GM, Griggs RC, Mendell JR, Moxley RT, Miller JP, et al. Clinical investigation of Duchenne muscular dystrophy. Interesting results in a trial of prednisone. Arch Neurol 1987;44:812–7.

There is a typo in the line ‘Only 9,43% of evaluated DMD patients were infected with COVID-19.’ as percent value should be written as ‘9.43%’ format that used throughout the manuscript.

We corrected.

Round 2

Reviewer 1 Report

acceptable